# Corporate digital transformation and carbon emission intensity: Empirical evidence from listed companies in China

**Pengyu Yang[1], Kejia Guo[1]\*, Jing Jia[2], Yulin Yin[3]**

1 College of Urban Economics and Public Administration, Capital University of Economics and Business, Beijing, China, 2 European School, Beijing International Studies University, Beijing, China, 3 College of Economics, Northwest Normal University, Lanzhou, China

\* gkjschoolemail@163.com

**Data Availability Statement:** All relevant data are within the manuscript and its Supporting Information files.

**Funding:** The author(s) received no specific funding for this work.

## Abstract

Digital transformation is crucial for sustainable development of enterprises and for addressing the conundrum of "efficiency and environment". Utilizing a dataset from A-share listed companies in China from 2007 to 2021, this paper investigates the direct impact, underlying mechanism and driving effect of enterprise digital transformation on carbon emission intensity. The findings reveal that: (1) At this stage, digital transformation in listed companies effectively reduces their carbon intensity, but the relationship between the two is not linear; instead, it exhibits a U-shaped trajectory, initially decreasing then increasing. (2) Analysis of mechanism indicates that costs associated with environmental governance and innovations in green technology serve as critical pathways through which corporate digital transformation influences carbon intensity. (3) The analysis of driving effect suggests that the digital transformation significantly curtails the carbon emission intensity of both upstream and downstream enterprises as well as those within the same industry and geographical region, through industrial linkage and the cohort effect. (4) Heterogeneity analysis elucidates that the digital transformation of enterprises in regions with stronger government environmental regulations has a markedly more pronounced effect on reducing the carbon emission intensity. Furthermore, the carbon emission reduction effect of digital transformation is more potent in capital-intensive and technology-intensive enterprises compared to labor-intensive enterprises. This paper offers valuable insights for fostering enterprise digital transformation and promoting green, low-carbon development aligned with the "dual-carbon" strategy.

## Introduction

Global climate change driven by substantial carbon dioxide emissions poses a formidable challenge to human sustainability and economic progression [1]. Addressing carbon emission reduction has become a pivotal and urgent aspect of sustainable development, which has received a great deal of attention from countries all over the world. As a prominent big country, China pledged at the 75th United Nations General Assembly to achieve a carbon peak by

**Competing interests:** The authors have declared that no competing interests exist.

2030 and carbon neutrality by 2060. However, unlike the developed countries in Europe and the United States that target a reduction in absolute carbon emissions, as a developing country, China must carefully balance the relationship between economic development and energy conservation and emission reduction. Achieving this balance, while ensuring economic growth stability and reducing the total amount of carbon emissions. Therefore, determining how to achieve the structural transformation of economic growth and to reduce the intensity of carbon emissions per unit of output value is pivotal in solving the dilemma between efficiency and the environment [2]. This transformation is also crucial for attaining the "dual carbon" goal and represents one of the most effective ways to achieve the goal of "double carbon". Enterprises, as the micro subjects of the market economy, constitute an important source of carbon emissions in China. Data show that the carbon dioxide emissions of the top 100 listed companies in China's total carbon emissions in 2022 will account for about half of the country's total carbon emissions [3]. Undoubtedly, these enterprises are the key object and basic unit of carbon emission reduction tasks, making it imperative to solve the carbon emission problem in the micro production field.

The rapid integration of digital technology into production processes has gradually become an important strategic choice for fostering enterprise growth and transformation. On the one hand, digital transformation mitigates information asymmetries, catalyzes changes in production modalities through technology, and enhances operational the improvement of enterprise production efficiency [4]. On the other hand, digital technology can help enterprises to extend beyond traditional production boundary, solve the productivity paradox problem, and optimize the allocation enterprise resource towards improved economic output [5]. Therefore, digital technology is expected to empower enterprises to achieve the reduction of carbon emission intensity. In recent years, policy directives from the Party Central Committee, under the leadership of Comrade Xi Jinping, emphasize the critical role of the deep integration of digitization and greening, and in February 2024, the Ministry of Industry and Information Technology and seven other departments jointly issued the "Guiding Opinions on Accelerating the Promotion of Greening Development of Manufacturing Industry," focusing on "giving full play to the enabling role of digital technology in promoting resource efficiency, environmental efficiency, management effectiveness, etc., and accelerating the synergistic transformation of the production mode of digitalization and greening". Digital greening synergistic transformation", which points out the direction for digital technology to empower carbon emission reduction. From the theoretical perspective, digital transformation will expand the production possibilities of enterprises, facilitating a more reasonable input-output ratio. From the practical level, it integrates digital transformation represents the deep integration of digital technology with the operational, managerial, and strategic facets of enterprises, which will help reconstruct the operation and management mode and business resources of enterprises, optimize the carbon emission frameworks, and strengthen the control of carbon production and capture from the output side. Consequently, this will achieve a balance between output growth and carbon emission reduction, thereby streamlining the carbon emission structure of enterprises, augmenting the management of carbon production and capture from the output side, and reducing carbon intensity by combining output growth and carbon emission reduction. Therefore, we focus on the microeconomic subject of enterprises, and scientifically evaluates the effect and mechanism of the digital transformation of enterprises on carbon emission intensity, which is not only a critical step towards advancing the green and low-carbon transformation of the economy, but also a vital requirement to ensure that our country achieves the "dual-carbon" goal and the high-quality development of the economy on schedule.

To the aim, we first established a theoretical analytical framework to assess the impact of corporate digital transformation on carbon intensity from three dimensions: direct impact,

mechanisms of action, and driving effects. Subsequently, we conducted empirical tests using panel data from China's A-share listed companies spanning 2007 to 2021, employing a two-way fixed effects model. Finally, we analyzed the heterogeneity of corporate digital transformation's effect on carbon intensity, considering the internal and external environments in which these corporations operate. Our findings indicated that the relationship between enterprise digital transformation and carbon intensity exhibits a U-shaped characteristic, initially decreasing before subsequently increasing. Currently, the digital transformation of listed companies effectively reduces their carbon intensity, with environmental governance costs and green technology innovation serving as critical transmission mechanisms. Furthermore, we observed that corporate digital transformation can not only reduce the carbon emission intensity of upstream and downstream enterprises through industrial linkage effects but also promote carbon emission reductions among enterprises within the same industry and region through cohort effects. Heterogeneity analysis revealed that the carbon emission reduction effect of corporate digital transformation is more pronounced in regions with stronger local government environmental regulations and higher capital and technology intensity.

In comparison to existing literature, this study presents its contributions primarily in three aspects. First, while existing research predominantly examines the impact of digitization on carbon emissions at the macro level [6, 7], there is a notable scarcity of relevant studies at the micro-enterprise level. Furthermore, the mechanisms through which digital transformation influences carbon emission intensity remain unclear. This paper addresses this issue by concentrating on the impact and mechanisms of digital transformation on carbon intensity within micro-production domains. It empirically assesses the carbon emission reduction performance associated with enterprise digital transformation using data from listed companies, thereby enriching the understanding of the drivers of enterprise carbon performance and providing new micro-empirical evidence for digitization-enabled green and low-carbon development. Second, existing literature predominantly adopt an aggregate perspective to analyze the carbon emission effect of digital transformation [8], which struggles to capture the interplay between social resource inputs and carbon outputs. Consequently, this study investigates the environmental performance of enterprise digital transformation from the carbon intensity viewpoint, which more accurately reflects the efficiency of energy use in economic activities and the enterprise's ability for carbon reduction. This approach expands the research on the environmental effects of enterprise digitalization. Third, scholars have increasingly focused on the spatial spillover effects of digital transformation on carbon emission reduction [9]. However, there has been limited exploration from the perspective of the supply chain within the industrial chain. This study explores the driving effect of enterprise digital transformation on carbon emission intensity through the lenses of industrial correlation effects and cohort effects, aiming to provide a more scientific and systematic assessment of the carbon performance following enterprise digital transformation.

The remainder of this paper is organized as follows: The second section presents a literature review of pertinent studies regarding the impact of digital transformation on carbon emissions. The third section establishes a theoretical framework that examines the influence of digital transformation on carbon intensity through three dimensions: direct impact, indirect impact, and driving effect, and formulates the research hypotheses of this study. The fourth section details the models, variables, and data employed in the research. The fifth section discusses the empirical results across six dimensions: benchmark regression, endogeneity issues, robustness testing, mechanism testing, driving effect analysis, and heterogeneity analysis. Finally, the sixth section summarizes the findings and outlines the policy implications of the research.

## Literature review

Enterprise digital transformation refers to a comprehensive integration of digital technology with the enterprise's organizational structure, business model, and production model, leading to a transformative overhaul of the enterprise's operation mode, decision-making mode, and organizational strategy [10]. The literature shows that on the micro level, predominantly examines the effects of digital transformation on human capital structure [11], innovation capacity [12] and production performance [13]. Conversely, studies on digital transformation impact on energy technology innovation [14] and environmental performance [15]. However, research on the relationship between digital transformation and carbon emissions remains limited, particularly concerning the relationship between digital transformation and carbon intensity of enterprises. Among them, Zhong and Ma (2022) found that digital transformation can significantly mitigate the level of corporate carbon risk [16], Jian (2024) paid attention to the fact that digital transformation can help to improve the level of corporate carbon disclosure [17], and Shang et al.'s (2023) study provided preliminary evidence for the impact of corporate digital transformation on carbon intensity [8]. However, studies adopting an aggregate perspective struggle to accurately depict the association between social resource inputs and carbon emission outputs.

On the macro level, previous studies have examined the relationship between digitalization and carbon emissions has been analyzed through various lenses such as ICT, Internet development and the digital economy, but conclusions remain inconclusive. Among them, some scholars assert that the application of ICT will be conducive to reducing resource consumption and improving regional energy efficiency [18], that the level of carbon emissions in cities are gradually decreasing with the deepening of the integration of digital technology and real industries [6], and Bai's (2023) study suggests that the digital economy not only directly mitigates carbon emissions, but also indirectly, by facilitating the transformation and upgrading of industries, contributes to carbon emissions [7]. Lin and Zhou (2021) found that the diffusion effects of digital technology also improves regional carbon emission performance to a certain extent [19]. Proponents of the neutral theory believe that there is a U-shaped relationship between digital construction and enterprise carbon performance, initially declining before improving [20], and suggest that the development of urban digitalization reaches a certain threshold to effectively foster collaborative emission reduction, transitioning the impact on carbon emissions from positive to negative [21]. Scholars who adhere to the detrimental theory have studied that the application of ICTs has exacerbated environmental degradation [22]. Avom (2020) observes that with the spread of ICTs, carbon dioxide emissions have increased in various countries in South Africa [23], primarily due to the accelerated development of digital machineries and the construction of digital infrastructures which significantly elevates consumption of resources, which in turn increases carbon emissions [24, 25].

To summarize, scholars domestically and internationally primarily focus on the impact of digitization on carbon emissions at the macro level, with less emphasis on the micro-level interplay between digitization and carbon emissions within individual enterprises. The relationship between digitization and the carbon intensity of enterprises remains underexplored, and the mechanisms by which digital transformation influences carbon emission intensity are not fully elucidated. Based on this, this paper primarily investigates the effect and mechanism of digital transformation on enterprises carbon intensity, and further examines the driving effect driven by the industrial linkage effect and the cohort effect.

## Theoretical analysis and research hypothesis

The essence of enterprise digital transformation lies in the utilization of digital technology to deeply infiltrate and revolutionize various aspects of the production process, business

management, strategic decision-making, and operational systems. This transformation aims to enhance market expansion opportunities while simultaneously achieving superior environmental and noteworthy economic performance [26]. In the following sections, we will delve into the influence of enterprise digital transformation on carbon emission intensity, focusing on three key areas: direct impact, mechanism of impact, and driving effect.

## Direct impact of digital transformation of enterprises on carbon intensity

At present, the role of digital transformation in diminishing carbon emission intensity is primarily manifested in the two aspects of improving enterprise production efficiency and reducing carbon dioxide emissions. Economically, digital transformation enables enterprises to more effectively integrate information flows, material flows and capital flows, thus realizing the multiplier effect of value, fundamentally overturning traditional modes of enterprises, gradually reduce the reliance on traditional factors of production, and foster a shift towards more efficient and intelligent production modes [27]. Furthermore, digital transformation significantly enhances the efficiency of information retrieval, transmission, exchange and management within the enterprise, optimizes the management structures, streamlines the approval process, and reduces the information asymmetry in production. In addition, digital transformation encourages enterprises to break away from traditional development models, comprehensively optimizes all aspects of enterprise production, and promotes the optimization and adjustment of the structure of production factors, thus enhancing the efficiency of resource factors utilization. From a carbon emission reduction perspective, firstly, digital transformation decreases the marginal costs of digital technology application, prompts enterprises to explore more diverse technology application scenarios, thus breaking the time and space constraints and information barriers in traditional resource allocation, thereby markedly enhancing energy use efficiency [28]. Secondly, enterprises can accurately monitor the spatial and temporal trajectories of waste pollutant gases through a digitalized waste pollutant gas early warning and monitoring platform, which informs the development of carbon capture and utilization strategies and effectively reduces carbon emissions intensity [29]. Finally, digital transformation can optimize the enterprise's "solid waste management" mode, using RFID technology and sensors, establishing a closed-loop management system for energy consumption information during the production and operation, and creating a circular production path of "energy-products-renewable energy", thus bolstering the enterprise's ability to reduce carbon emissions [30].

Digital transformation not only bolsters enterprise productivity of enterprises but also significantly lowers carbon emissions, thereby reducing the intensity of carbon dioxide emissions. However, in terms of dynamics, digital transformation represents a complex and extensive systematic project. During the transformation process, the replacement of equipment, the application of new technologies, technological exploration, and the energy rebound effect may result in a non-linear relationship between the role of digital transformation and carbon intensity. At the initial stage, digital transformation often involves substantial fixed assets repurchases and hardware investments; the short-term procurement of new equipment and disposal of old equipment may increase carbon emissions [31]. And due to the surge in data processing and storage demands could escalate energy consumption at data centers. The demand for digital talents and staff training at the outset of digital transformation may temporarily decrease resource utilization efficiency, rendering the carbon reduction benefits less visible at this stage, with digital inputs will have a limited role in improving output efficiency [32]. In the middle stage, as technology advances and optimizes, the management of digital resources will be more reasonable, and the efficiency of the digital center will be further

realized, thus reducing carbon emissions [33]. The familiarization of employees with digital technologies and processes not only reduces labor costs, but also optimizes production processes [34], fully demonstrating the carbon performance of enterprise digital transformation at this stage [35]. In the later stage, as the application of digital technology reduces the unit cost of product production, which continues to decline. This prompts enterprises to increase output levels to maximize profits and compensate for the substantial investments made in digital transformation, thus generating additional energy demand, and carbon emissions in production—the rebound effect of carbon emissions [36]. Based on the above analysis, we propose the following research hypothesis:

Hypothesis 1(H1): Digital transformation reduces corporate carbon intensity, but there may be a U-shaped relationship between digital transformation and carbon emission intensity.

## The effect mechanism of digital transformation on carbon emission intensity

Digital transformation significantly influences corporate carbon intensity, not only directly, but also indirectly by reducing environmental governance costs and fostering green technology innovation.

From the perspective of "cost reduction" in enterprise digital transformation, studies have assessed the impact on enterprise operating costs [37]. However, the effects on environmental governance costs have received less attention. The "Porter Hypothesis" suggests that moderate environmental regulations compel firms to enhance their green innovation capacity, this is because the "innovation compensation effect" of environmental regulation is greater than the "cost crowding out effect" on enterprises. In the context of increasing environmental regulation, business managers are increasingly recognizing the importance of integrating environmental factors into their strategic decision-making. Digital transformation emerges as a crucial facilitator in this integration. Through the comprehensive application of digital technology, the "wisdom +" model is progressively penetrating various facets of enterprise management, reshaping the management and decision-making processes. Specifically, the advancement in digital technology can help enterprises to conduct deeper analyses of their energy intensity and carbon emissions in production and operation. By integrating digital technology and enterprise energy management system, businesses can establish intelligent energy management systems. These systems facilitate more precise predictions and management of actual demand for energy, achieving the total amount of control and intensity control, thereby reducing the environmental management costs associated with emissions and enhancing cost-effectiveness of enterprises. In this way, the environmental management costs caused by sewage discharge can be reduced and the cost-effectiveness of the enterprise can be improved. In addition, the reduction of environmental management costs may encourage enterprises to invest more in energy-saving and emission-reducing green technologies. These technologies typically enhance the efficiency of energy utilization, reduce resource consumption and curtail waste emissions, and thus reduce the carbon emission intensity of enterprises.

From the perspective of the "efficiency" of enterprise digital transformation, the deepened application of digital technology improves technical performance, optimizes the innovation process, and fosters a constructive cycle of positive feedback. This is achieved through the creation of favorable environment for enterprises and the improvement of the level of green technological innovation of enterprises [38], thereby supporting the sustainable development of enterprises. development to provide green support. The process of green technology innovation often involves multiple iterations of trial and error, optimized through the capabilities of

digital platform that offer efficient data collection and immediate feedback. By establishing digital simulation platforms, enterprises can enhance their green technology research and development and innovation processes, broadening their internal resource allocation for innovation, so as to ensure that the enterprise can be efficient research and development to achieve predictable green technology innovation. The enhancement of green technology innovation capability can further refine the production processes and energy configurations of enterprises, reducing reliance on traditional energy and promoting the transformation of the energy structure to a more environmentally friendly direction, but also greatly improves the efficiency of energy utilization [39], thus realizing the reduction of the intensity of corporate pollution emissions. Based on the analysis provided, we propose the following research hypothesis:

Hypothesis 2(H2): Corporate digital transformation will indirectly affect corporate carbon intensity through reductions in environmental governance costs and enhancements in green technology innovation.

## The driving effect of digital transformation on carbon emission intensity

Digital technologies, by virtue of its extensive penetration, is progressively enabling comprehensive technological innovation of enterprise production processes, operational management, and strategic decision-making. As a fundamental component of digital technology, data is radically transforming modes of production, living and working in human society, characterized by its openness and value generation [40]. Unlike traditional factors of production, data factors, possess non-competitive attributes and benefits from unlimited re-use. Together with the profound penetration characteristics of digital technology, it continually to expand the scope of scientific and technological activities, industrial boundaries, and the transformation of the results of the border, all-round amplification of the real economy's technological expansion space [41]. Thus, "Driving" traditional production methods through industrial linkages and peer competition.

First of all, the digital transformation of enterprises will drive the carbon emission reduction across the industrial chains and the upstream and downstream enterprises of supply chain through the industrial correlation effects. With the continuous expansion of the market scale and the deepening of the professional division of labor among industries, the connection between enterprises tighten, and the changes in the production mode and technology level of individual enterprises will inevitably produce industrial correlation effects with the core enterprises as the nodes [42], thus fostering low-carbon development across industrial chain and supply chain as a whole. On the one hand, the digital transformation of enterprises effectively mitigates information islands between enterprises within these chains, realizes point-to-point transmission of information exchange between enterprises, solves the issues of information asymmetry through precise information matching, which reduces unnecessary energy consumption. On the other hand, the deepening of digital technology application in the upstream industry will provide cleaner, greener intermediate products to downstream industry, which will effectively promote the downstream industries to realize the transformation of technological innovation and production mode by superimposing the catalytic "matching effect", thereby generating a positive carbon emission reduction effect among industries. In addition, due to its strong economic characteristics of permeability, synergy, substitutability and creativity, digital technology will promote the formation of multi-level product, technology, ecological and value networks among enterprises along the industrial chain and supply chain. This enhances the continuous extension and integration of the industrial chain, supply chain, value chain, and innovation chain within the economic system, and ultimately leading to a comprehensive

enhancement in carbon emission reduction capabilities across upstream and downstream industries.

Secondly, enterprise digital transformation substantially lowers carbon emissions through the peer effects. In addition to the constraints of its own resources, an enterprise's operation and production decisions are also influenced by the market environment, stakeholders in the same industry and region, and other factors. Digital transformation enables enterprises to swiftly capture market dynamics and rapidly adjust resource allocation, which is a critical strategy for securing market advantages in complex dynamic systems. When enterprises in an industry or region capitalize on market opportunities through the power of digital transformation, this initiates a series of "demonstration-imitation" responses, motivating competitors to undertake digital transformation in order to maintain their own competitive advantages. Driven by the cohort effect of digital transformation, the degree of enterprise digital transformation will be further deepened, thus further enhancing the improvement of carbon emission utilization, expanding the application scenarios of digital technology and further optimizing the carbon emission reduction methods such as "solid waste management", and improving the efficiency of enterprise energy use. Given the above analysis, we propose the following research hypothesis.

Hypothesis 3(H3): The digital transformation of enterprises can reduce the carbon emission intensity of enterprises within the same industry and region, as well as upstream and downstream of the industrial chain supply chain, through the industrial correlation effects and the peer effects.

In summary, we constructed a theoretical framework, as shown in Fig 1:

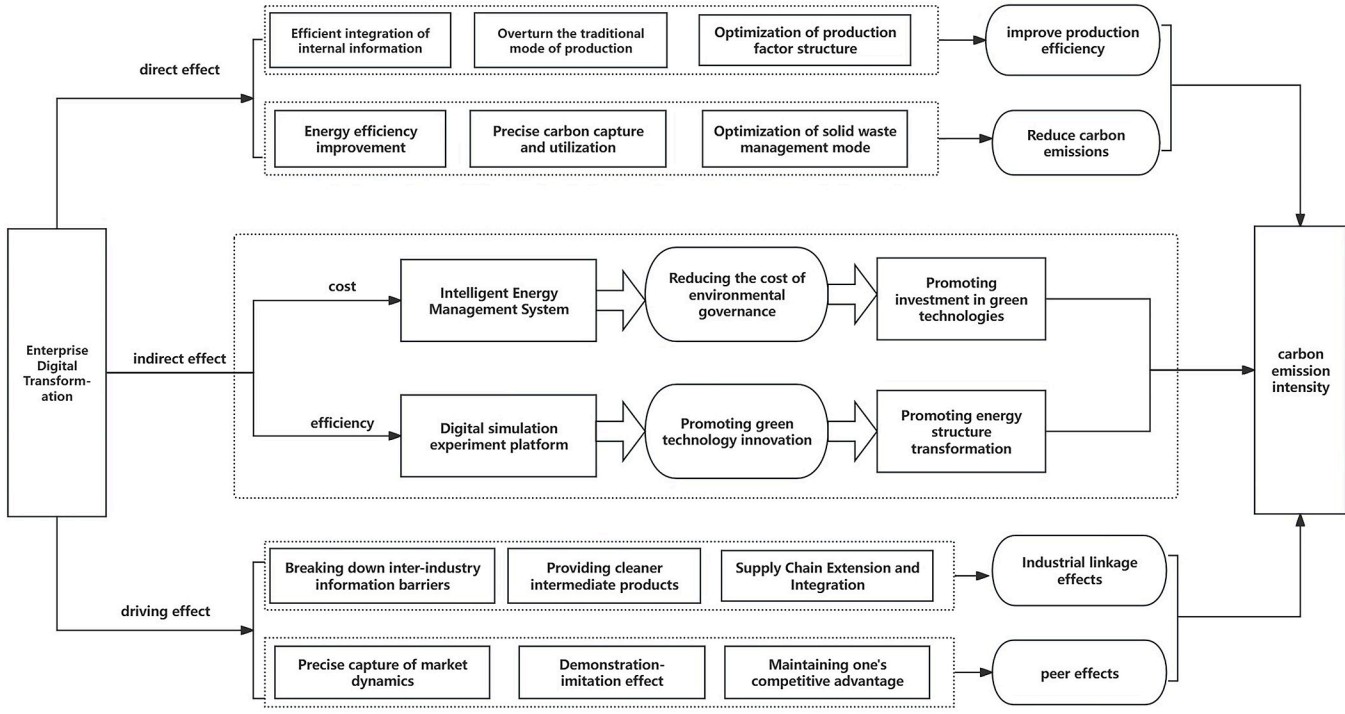

**Fig 1. The schematic diagram of the intrinsic mechanism of enterprise digital transformation affecting carbon intensity.**

## Empirical research design

### Models

This paper investigates the influence of digital transformation on the carbon emission intensity of enterprises at the micro level. In order to test the direct impact of the digital transformation of enterprises on the carbon emission intensity, the following benchmark model is established:

$$CEI_{it} = \beta_1 dig_{it} + \beta X_{it} + \alpha_i + \theta_i + \mu_{it} \tag{1}$$

Where $i$ denotes province and $t$ denotes year. *CEI* and *dig* denote the carbon emission intensity and Level of digital transformation, respectively. *X* is a set of control variables that affect the Carbon Emission Intensity of enterprises. $\alpha_i$, $\theta_i$, and $\mu_{it}$ represent individual fixed effect, year fixed effect and random disturbance term, respectively.

Secondly, building on previous theoretical analysis, he digital transformation of enterprises may have an indirect impact on the carbon emission intensity through environmental governance costs and green technological innovation. We employ a mediating-effect model to examine this transmission mechanism. Referring to Wang (2022) [43], the following model is constructed:

$$M_{it} = \beta_0 + \beta_1 dig_{it} + \alpha X_{it} + \mu_i + \eta_t + \varepsilon_{it} \tag{2}$$

$$CEI_{it} = \beta_0 + \beta_1 dig_{it} + \beta_2 dig_{it} * M_{it} + \alpha X_{it} + \mu_i + \eta_t + \varepsilon_{it} \tag{3}$$

Where M represents the mechanism variables, including corporate environmental governance costs (EC) and green technology innovation (GII), and the other variables have the same meaning as in Eq (1).

### Variables

1. The explained variable. Carbon Emission Intensity (CEI). Based on the methodology of Lv (2024) [44], carbon intensity is quantified as the ratio of carbon dioxide emissions to the main business income of enterprises. Carbon intensity = carbon dioxide emissions × 108 / enterprise main business income. Among them, enterprise carbon emissions = (industry energy consumption × carbon dioxide conversion coefficient × enterprise business cost)/ industry business cost, carbon dioxide get conversion coefficient using Xiamen Energy Conservation Center's standard 2.493. The treatment of missing data refers to the practice of Wu et al. (2022) [45], which utilizes the ratio of the industrial output value of listed companies to the gross industrial product of prefecture-level cities as the energy consumption coefficient. This coefficient is then applied to estimate the $CO_2$ emissions of prefecture-level cities to approximate the $CO_2$ emissions of listed companies.

2. The core explanatory variable. Digital transformation (dig). Referring to the practice of Wu et al. (2021) [46], a text mining method is used to extract keywords from annual reports of listed companies, covering five dimensions of digital technology application, artificial intelligence, big data, blockchain, and cloud computing. This information forms a digital transformation lexicon, quantifying the extent of digital transformation of listed companies based on the ratio of the phrases to the total number of words in the annual reports.

3. The mechanism variables. (1) Environmental governance cost: Drawing on the study by Liu et al. (2022) [47], this is represented by the sewage charges of enterprises are used to characterize the environmental governance of enterprises (EC). (2) Green Technological Innovation. (GII): we refer to Xu et al. (2020) [48], this is measured by the number of green

**Table 1. Descriptive statistics of variables.**

| Variables | N | Mean | Std. Dev. | Min | Max |
|---|---|---|---|---|---|
| CEI | 27196 | 159.436 | 263.055 | 0 | 29275.31 |
| Dig | 27196 | 2.909 | 7.741 | 0 | 91 |
| EC | 27196 | 612.002 | 743.644 | 44.336 | 2893.145 |
| GII | 27196 | 1.922 | 6.614 | 0 | 64 |
| Roa | 27196 | 4.229 | 5.944 | -36.514 | 25.441 |
| state | 27196 | 0.151 | 0.358 | 0 | 1 |
| bsize | 27196 | 8.663 | 1.698 | 5 | 15 |
| dshare | 27196 | 37.309 | 5.301 | 22.22 | 57.14 |
| Bm | 27196 | 0.626 | 0.242 | 0.095 | 1.236 |
| dual | 27196 | 0.273 | 0.445 | 0 | 1 |
| Age | 27196 | 10.364 | 7.308 | 1 | 29 |

patent applications filed by listed companies in the year, in order to guarantee the smoothness of the data and to avoid the influence of the value of 0, the number of patent applications plus 1 is taken in the form of logarithm.

4. The control variables. For weakening the possible the estimation bias caused by omitted variables in the models, the following control variables are included. (1) Net profit margin on total assets (roa), measured by the profitability of the enterprise to its average total assets, multiplied by 100. This measure aims to align with the magnitude of the data for the explanatory variables in this paper. (2) Nature of controlling shareholders (state), Indicates whether the controlling shareholder of the listed company is state-owned or state-controlled is 1, otherwise it is 0. (3) board size (bsize), measured by natural logarithm of the number of board members. (4) Percentage of independent directors (dshare), measured by number of independent directors to number of board of directors. (5) Book-to-market ratio (bm), expressed as the ratio of listed company share price per share to net assets. (6) The integration of two positions (dual), if the chairman and general manager (or CEO) of a listed company are held by the same person as 1, otherwise it is 0. (7) Age of business (age), measured by the year of the listed company minus the year of inception and taking the logarithm.

## Data sources and descriptive statistics

Energy consumption data of the industry in this paper is sourced from China Energy Statistical Yearbook. The industry operating cost data is from the China Industrial Economic Statistics Yearbook. The data of listed companies come from the Cathay Pacific database, covering the period from 2007 to 2021. According to the industry code, we excluded sectors like real estate and financial, ST, *ST, PT and other types of companies, the missing data of key indicators are also deleted. Considering the matching problem of the data, we deleted the data of some listed companies that failed to match, and performed a 1% bilateral tail reduction on all continuous variables. Finally, a final sample of 3,296 companies were obtained in 2021, with a total sample size of 27,196. Table 1 presents the descriptive statistical results of the variables.

## Results and discussion

### Benchmark regression

To ensure the accuracy of the estimation results, we examined the multicollinearity among variables prior to conducting the baseline regression. The Variance Inflation Factor (VIF) of

**Table 2. Benchmark regression results.**

| variable | (1) | (2) | (3) | (4) | (5) | (6) |
|---|---|---|---|---|---|---|
| | CEI | CEI | CEI | CEI | CEI | CEI |
| dig | -0.121** | -0.103 | -0.223*** | -0.166*** | -0.186*** | -0.395*** |
| | [0.047] | [0.066] | [0.071] | [0.051] | [0.062] | [0.134] |
| $dig^2$ | | | | | | 0.004** |
| | | | | | | [0.002] |
| roa | | -1.123 | -2.356** | -2.342*** | -2.283*** | -2.281*** |
| | | [0.907] | [0.900] | [0.793] | [0.755] | [0.755] |
| state | | -6.248 | -2.71 | -2.779 | -2.42 | -2.387 |
| | | [5.025] | [3.809] | [3.750] | [4.739] | [4.748] |
| bsize | | 3.789*** | 0.839 | 2.026 | 2.090* | 2.101* |
| | | [1.137] | [1.798] | [1.479] | [1.143] | [1.144] |
| dshare | | 1.246*** | 0.503 | 0.545* | 0.681*** | 0.681*** |
| | | [0.254] | [0.317] | [0.289] | [0.234] | [0.236] |
| bm | | -1.447 | -88.002** | -90.486** | -85.622** | -85.704** |
| | | [13.275] | [40.327] | [40.134] | [34.862] | [34.886] |
| dual | | -4.936* | -4.869* | -1.59 | -1.905 | -1.913 |
| | | [2.821] | [2.583] | [2.186] | [1.172] | [1.178] |
| age | | 0.065 | -16.265 | -14.867 | -19.576 | -19.649 |
| | | [0.242] | [9.738] | [8.744] | [12.966] | [13.023] |
| u-test | | | | | | 2.03** |
| Year FE | yes | yes | yes | yes | yes | Yes |
| Individual FE | no | no | yes | yes | yes | Yes |
| City FE | no | no | no | yes | yes | Yes |
| Industry FE | no | no | no | no | yes | Yes |
| N | 27196 | 27196 | 26821 | 26819 | 26819 | 26819 |
| $R^2$ | 0.001 | 0.003 | 0.183 | 0.189 | 0.191 | 0.191 |

Note

***, ** and * indicate significance level of 1%, 5%and 10%, respectively.

each variable is less than 1.41, significantly lower than 10, suggesting minimal multicollinearity. So, it can be basically concluded that there is no serious multicollinearity between the variables, and the regression model estimation can be carried out further. To capture the impact of digital transformation on the carbon emission reduction performance of listed companies' digital transformation, referring to the treatment of Liu et al. (2023) [15], Table 2 presents the baseline results obtained by adopting a progressive regression strategy. Among them, column (1) shows the regression outcomes excluding control variables, and columns (2) through (5) correspond to the results with the progressive inclusion of control variables, individual effects, region effects, and industry effects, respectively. In order to improve the computational efficiency, this paper adopts the estimation command with high-dimensional fixed effects. After controlling for the individual effects, the regression results converge in 8 iterations and 375 sample observations are automatically excluded, resulting in a reduced samples in column (3). And in column (4) of Table 2, due to the inclusion of the area estimation effect, the regression results converge in 132 iterations and 377 sample observations are automatically removed, indicating further shrinkage of the regression sample. Regardless of the regression strategy, the effect of enterprise digital transformation on enterprise carbon emission intensity is negative, with most results achieving statistical significance at the significant 1% level. This substantiates

the first part of Hypothesis 1, affirming that an increase in the degree of enterprise digital transformation significantly reduces its carbon emission intensity. This conclusion aligns with the findings of Zhong and Ma (2022) [16] and Ma and Yang (2023) [49], which suggest that the digital transformation of listed companies yields dual advantages of enhanced main business revenue and reduced carbon emissions.

We explored the nonlinear impact of enterprise digital transformation on carbon emission intensity by including the square term of the explanatory variables (dig2) is included in Eq (1) and conducting further regression. The results of column (6) reveal that the impact of listed companies' digital transformation on carbon emission is still significantly negative after the inclusion of the square term, with the coefficient of the quadratic term is significantly positive, which is a nonlinear characteristic. There may be a "U-shaped" relationship, attributable to the lagging economic benefits of enterprise digital transformation. In the initial stage of enterprise digital transformation, subject to employee proficiency, digital infrastructure construction, digital application scene construction and a series of impacts, digital transformation of the release of green efficiency is inadequate. However, as digital integration into production and management processes deepens, the benefits become increasingly evident. The effect of economies of scale continues to highlight, the cost per unit of carbon emission reduction declines, and green, low-carbon technologies mature. Consequently, enterprise experience significantly enhanced carbon and emission reduction outcomes. Further, the regression results were analyzed according to Haans' (2016) [50] U-curve test to assess if they meet the criteria for a U-shaped (inverted U-shaped) curve. The results show that the coefficient of the primary term of the digital construction level is significantly negative (-0.395***), while the secondary term's coefficient is significantly positive (0.004***), which satisfies the first determination condition; the digital construction level in this paper takes the value range of [0,91], and the slope at the left endpoint is negative (k1 = -0.395), and the slope at the right endpoint is positive (k2 = -0.395 + 2 × 0.004 × 91 = 0.333), satisfying the second condition; after calculation, the inflection point is x1 = —(-0.395)/(2 × 0.004) = 49.375, which falls within the range of values of the enterprise digital transformation and meets the third condition. As a result, this confirms a U-shaped relationship between the digital transformation of enterprises and carbon intensity, which first decreases and then increases, thereby validating Hypothesis 1 (H1). In addition, the U-test of the U-shaped conclusion of this paper is supported by using Stata software, The results of the U-shape test show that the t-value is 3.86 and the p-value is 0.000, which passes the U-shape test. This pattern echoes the findings of Xiao (2023) [20], who initially explored the nonlinear relationship between digitization level and enterprise carbon performance based on a sample of the provincial level in China. We further extended the correlation analysis by employing a more micro-specific sample of firms.

## Responses to endogeneity problems

Benchmark regression models may encounter endogeneity problems. On the one hand, digital transformation, as an essential strategy for the green and low-carbon development of listed companies, might exhibit a mutually reinforcing effect between the two. In practice, achieving corporate carbon emission reduction targets often necessitate the overhaul of low-carbon equipment and the updating of green technology. Companies with superior carbon emission performance may initially adopt digital technology and the platform economy, furthering digital maturity and potentially leading to bidirectional causality, which could skew the benchmark results. On the other hand, although in the design of the benchmark regression model, the relevant factors affecting the carbon emission performance of listed companies' digital transformation have been controlling for year, individual, region and industry, it is difficult to

**Table 3. Endogenous treatment.**

| variable | (1) | (2) | (3) | (4) | (5) | (6) |
|---|---|---|---|---|---|---|
| | dig | CEI | dig | CEI | CEI | CEI |
| IV1 | 0.021*** | | | | | |
| | [0.003] | | | | | |
| dig | | -7.219** | | -1.695*** | | |
| | | [3.667] | | [0.642] | | |
| IV2 | | | 0.946*** | | | |
| | | | [0.015] | | | |
| di*dt | | | | | -5.430*** | |
| | | | | | [1.880] | |
| di*dt*dig | | | | | | -0.186*** |
| | | | | | | [0.062] |
| _cons | 4.877*** | 123.339*** | 0.563 | 95.863*** | | |
| | [0.601] | [27.211] | [0.534] | [18.607] | | |
| N | 25307 | 25307 | 27180 | 27180 | 26819 | 26819 |
| LM statistic | 6.017*** | | 3.677** | | | |
| Wald Fstatistic | 62.216 | | 3728.592 | | | |

Note

***, ** and * indicate significance level of 1%, 5%and 10%, respectively.

guarantee that unobservable factors affecting the carbon emission performance of listed companies may still exist and be included, and thus the possibility of omitted variables still exists, and therefore, to address these issues, we adopted two approaches.

Firstly, inspired by Huang et al. (2019) [51], we utilized the number of post offices per million people in each prefecture-level city in 1984 is used as an instrumental variable. Considering that the number of post offices per million people in the city in 1984 is a historical variable, which may be difficult to identify during the estimation of the fixed-effects models [52], we draw on the study of Nunn and Qian (2014) [53], by constructing an interaction term between the number of post offices in the city in 1984 (related to individual changes) and the more exogenous national Internet penetration rate (related to time), is constructed as the instrumental variable (IV1) in this paper and empirically tested using two-stage least squares estimation. The regression results from Table 3 reveal that, in the first-stage results of the two-stage regression in Column (1), the density of post offices per million people in 1984 significantly promotes digital transformation of firms. In addition, the results of instrumental variable test confirm the absence of issues such as under-identification and weak instrumental variables, and the instrumental variable selection is valid. On this basis, the results of the second stage of Column (2) show that the digital transformation of enterprises consistently reduces the carbon emission intensity of enterprises, aligning with the conclusion of the benchmark regression.

Secondly, drawing on the research frameworks proposed by Xiao et al. (2022) [11] and Li et al. (2022) [54], this study utilizes the average degree of digital transformation among firms within the same industry and city is used as an instrumental variable (IV2) for endogeneity testing. Theoretically, within the same city and the same industry, the average digital transformation of all enterprises other than this enterprise has an obvious driving effect on the digital transformation of this enterprise, which satisfies the relevance condition; Moreover, the degree of digital transformation of enterprises within the same industry in this region does not affect this firm's carbon emission intensity, thus meeting the exogeneity condition. The regression

presented in Table 3, columns (3)-columns (4), demonstrate that the coefficient of IV2 in the first-stage regression is significant at the 1% level, confirming the validity of the under-identification and weak instrumental variables test, which supports the reasonableness of the instrumental variables selected in this paper. The coefficient of dig in the second stage regression remains significantly negative at the 1% level, substantiating the robustness and reliability of the paper remain robust and reliable with the use of instrumental variables.

Thirdly, Reference on Wu Fei et al. (2021) [46], the double-difference method was employed to assess the precise impact of digital transformation on corporate carbon emission reduction of listed companies. Digital transformation activities were carried out as a criterion to differentiate between the experimental group and the control group, thereby minimizing inherent individual differences and reducing estimation bias caused by the trend of events. Referring to Liu et al. (2022) [55], Eq (4) and Eq (5) were constructed to assess the impact of digital transformation on the carbon intensity of enterprises.

$$\mathbf{CEI}_{it} = \boldsymbol{\beta}_1(\boldsymbol{dt}_{it} \times \boldsymbol{du}_{it}) + \boldsymbol{\beta X}_{it} + \boldsymbol{\alpha}_i + \boldsymbol{\theta}_t + \boldsymbol{\mu}_{it} \tag{4}$$

$$\mathbf{CEI}_{it} = \boldsymbol{\beta}_1(\boldsymbol{dt}_{it} \times \boldsymbol{du}_{it} \times \boldsymbol{dig}_{it}) + \boldsymbol{\beta X}_{it} + \boldsymbol{\alpha}_i + \boldsymbol{\theta}_t + \boldsymbol{\mu}_{it} \tag{5}$$

In Eq (4), if the enterprise has undertaken digital transformation during the research period, $du_{it}$ takes 1, otherwise it takes 0, In the year when the enterprise carries out digital transformation and the year after that $dt_{it}$ takes 1, and vice versa takes 0. The coefficient of $dt_{it} \times du_{it}$ reflects the change of carbon emission intensity of enterprises before and after digital transformation. In Eq (5) further incorporates the degree of digital transformation of listed companies, and the coefficient of $dt_{it} \times du_{it} \times dig_{it}$ eflects the impact of the degree of digital transformation of companies on carbon intensity.

The regression results, as illustrated in Table 3, indicate that using digital transformation as a pilot policy to alleviate the endogeneity significantly enhances enterprise carbon emission reduction performance. This indicates that the enterprise's digital transformation behavior will accelerate the integration of digital technology across various aspects of enterprise production, such as division of labor, production collaboration, overall productivity, thereby optimizing production efficiency, reducing the loss of factors of production, enhancing the efficiency of energy use, and supporting the enterprise's carbon emission reduction.

## Robustness tests

To ensure the validity of the benchmark regression results, we implemented the following three approaches for robustness testing. First of all, considering that the text analysis method employed in the constructing the degree of digital transformation of listed companies results in numerous zero values, these zeros were retained in line with the principle of maintaining the integrity of the sample. However, to avoid potential bias in the estimation results of the digital transformation on the carbon emission reduction, the samples with a zero digital transformation index are excluded, and the benchmark regression estimation is re-run. The results shown in column (1) of Table 4, confirm that the negative impact of enterprise digital transformation on carbon emission intensity remains significant at the 5% significance test, further substantiating the robustness of the conclusions of the benchmark regression in this paper. Second, the 2015 sample is excluded due to the abnormal fluctuations in the share prices, compared with other years. During such period, listed companies may boost investor confidence by amplifying the disclosure of terms related to digital transformation in their annual reports. This, in turn, potentially elevates their share prices, thus affecting the estimation outcomes discussed of this paper. Therefore, the regression analysis is re-run after omitting the estimation

**Table 4. The robustness test.**

| variable | (1) | (2) | (3) |
|---|---|---|---|
| | CEI | CEI | CEI |
| dig | -0.137*** | -0.146** | -0.192*** |
| | [0.045] | [0.061] | [0.057] |
| city*year | | | -0.000** |
| | | | [0.000] |
| Year FE | YES | YES | YES |
| Individual FE | YES | YES | YES |
| City FE | YES | YES | YES |
| Industry FE | YES | YES | YES |
| N | 10380 | 25242 | 27003 |
| $R^2$ | 0.288 | 0.190 | 0.191 |

Note

***, ** and * indicate significance level of 1%, 5%and 10%, respectively.

sample of 2015, with results displayed in column (2) of Table 4. It is evident that the regression results concerning firms' digital transformation remain significant, confirming that the foundational regression results of this paper are robust, having accounted for sample outliers and possible confounding variables. Finally, by integrating cross-multiplication fixed effects of prefecture level city and year, and aligning with Chen (2020) [56], the regression is re-run by incorporating the cross-multiplier interaction between city and year in the baseline regression model. and the results are shown in column (3) of Table 4, indicate that the estimated coefficients and directional consistency of the core explanatory variable remain aligned with the benchmark regression, further reinforcing the validity of the benchmark regression results.

## Analysis of impact mechanisms

According to the previous theoretical analysis, the digital transformation of listed companies is posited to influence the carbon emission intensity of enterprises by reducing the cost of environmental governance and promoting green technological innovations. The regression analysis, employing Eqs (2) and (3), is executed with results delineated in Table 5. The estimation results in column (1) show that at this stage, the digital transformation of listed companies increases the enterprise's sewage costs, with no evident cost reductions in environmental governance manifesting. This may be due to the nascent stage of transformation where the extensive deployment of a large number of digital technologies elevates corporate carbon emissions. Additionally, the complexity of the digital systems and management changes during the early stage of implementation, potentially due to improper implementation of technology or poor management, may lead to poor environmental governance, subsequently raising governance costs. However, the estimation results incorporating a quadratic term of digital transformation in column (2) show that this improvement process presents non-linear characteristics, that is, as the degree of digitization intensifies, the enterprise's cost reduction effect will gradually emerge. This is likely due to the profound application of digital technology, big data and machine learning enabling more accurate monitoring and optimization of the enterprise's energy consumption and carbon emissions through big data algorithms to achieve further optimization. Optimization can also be achieved through the Internet of Things (IoT) technology, which enables enterprises to facilitate remote equipment with low energy consumption, thereby reducing their environmental management costs. Column (3) illustrates the regression

**Table 5. Mechanistic testing.**

| variable | (1) | (2) | (3) | (4) | (5) | (6) |
|---|---|---|---|---|---|---|
| | EC | EC | CEI | GII | GII | CEI |
| dig | 2.169** | 5.523*** | -0.345*** | 0.027*** | 0.075** | -0.224*** |
| | [0.981] | [1.375] | [0.079] | [0.009] | [0.027] | [0.055] |
| $dig^2$ | | -0.062*** | | | -0.001*** | |
| | | [0.011] | | | [0.000] | |
| EC*dig | | | 0.000*** | | | |
| | | | [0.000] | | | |
| GII*dig | | | | | | 0.011*** |
| | | | | | | [0.003] |
| u-test | | 4.02*** | | | 2.73*** | |
| Sobel—p | | | 0.000 | | | 0.000 |
| Percentage of intermediary effects | | | 23.0% | | | 22.5% |
| Control variables | YES | YES | YES | YES | YES | YES |
| Year FE | YES | YES | YES | YES | YES | YES |
| Individual FE | YES | YES | YES | YES | YES | YES |
| City FE | YES | YES | YES | YES | YES | YES |
| Industry FE | YES | YES | YES | YES | YES | YES |
| N | 26819 | 26819 | 26819 | 26819 | 26819 | 26819 |
| $R^2$ | 0.867 | 0.868 | 0.191 | 0.649 | 0.65 | 0.191 |

Note

***, ** and * indicate significance level of 1%, 5% and 10%, respectively.

results post-integration of the interaction term between enterprise digital transformation and environmental governance costs, and noting that the impact of digital transformation on enterprise carbon intensity is still significantly negative, whereas the estimated coefficient of the cross-multiplier term appears significantly positive at 1% significance level. This suggests an amplifying the inhibitory effect of the digital transformation of listed companies on carbon intensity as environmental governance costs decrease, offering firms greater resources and incentives to promote digital transformation and the application of green technologies. Although the digital transformation of enterprises at this stage has not significantly reduced the cost of corporate environmental governance, the continued integration of digital technology and the production processes is expected to reduce these costs. This reduction will significantly contribute to the reduction of corporate carbon intensity.

Columns (4)-(6) elucidate the impact of enterprises' digital transformation on carbon intensity through green technology innovation. Results from column (4) show that the digital transformation of listed companies at this stage has a significant positive impact on their green technology innovation at the 1% statistical significance level. This indicates that advanced digital applications enhance the transformation of listed companies in the direction of data-driven, algorithmic optimization and intelligent decision-making processes. This notably reduces the research and development cycle of green and low-carbon technologies, boosting the level of green technology innovation. Column (5) introduces the quadratic term of digital transformation, indicating that the impact of digital transformation on green technological innovation follows an inverted "U" trend—initially stimulating and then weakening. As digital transformation deepens, enterprises may face technological bottlenecks, such as the difficulty of the existing digital technology to meet the demand for higher-level green technological innovation, which leads to the late stage of green technological innovation. The results in column (6),

which incorporate the interaction of digital transformation and green technology innovation, show that as the level of green technology innovation increases, the inhibitory effect of digital transformation on carbon intensity notably diminishes. This is attributable to the fact that green technology innovation itself can significantly reduce the carbon intensity of enterprises [38]. As the level of green technological innovation increases, firms may rely more on techno-logical innovation, rather than digital transformation, to reduce carbon intensity. This substi-tution effect diminishes the role of digital transformation in reducing carbon intensity.

In order to ensure the credibility of the results in Table 5, this paper takes two routes to re-test: firstly, it tested the Sobel value of the indirect effect reveals the p-value of Sobel's test all passed the statistical test at the 1% level, firmly rejecting the original hypothesis, so it can be assumed that both the environmental governance cost and green technology innovation have significant mediation effect, with mediation effect share is 23% and 22.5%, respectively. Sec-ond, following the methodology proposed by preacher and Hayes (2008), the nonparametric percentile Bootstrap method test was conducted with a sampling number of 1000. The 95% confidence intervals (adjusted for bias) for the coefficients of the indirect effects were calcu-lated. The results showed that the Bootstrap test confidence intervals for environmental gover-nance costs and green technology innovation were [-0.790, -0.168] and [-0.698, -0.250] respectively. Both intervals had negative upper and lower limits, indicating a significant media-tion effect for both variables.

## Analysis of the driving effect

Many scholars have substantiated the spatial driving effect of enterprise digital transformation, and this paper primarily investigates the carbon emission performance of listed companies' digital transformation within the industry chain and its impact facilitated by the peer effect.

First of all, to examine the industrial correlation effect of carbon emission reduction due to enterprise digital transformation, the following regression model is set:

$$dig_{it} = \beta_1 dig_{down/up_{it}} + \beta X_i + \alpha_i + \theta_i + \mu_{it} \tag{6}$$

$$CEI_{it} = \beta_2 dig_{down/up_{it}} + \beta X_i + \alpha_i + \theta_i + \mu_{it} \tag{7}$$

Where $\beta_1$ indicates the impact of the digital transformation within companies upstream or downstream of the industrial supply chain on the digital transformation of i- firms, $\beta_2$ captures the impact of digital transformation in downstream or upstream firms on the carbon intensity of i-firms. Referring to Tao et al. (2023) [57] for the measurement methodologies of industrial and supply chain, consider that the upstream enterprise (A) may serve multiple downstream customers (Q1, Q2, Q3) in a certain year (2021), leading to the construction of A-Q1-2021, A-Q2–2021, A-Q3-2021 observations. This establishes the upstream and downstream enter-prise-annual data in this paper. Given data alignment challenges, this paper aggregates samples in the year 2007–2021, with 5236 downstream observation samples and 2022 upstream obser-vation samples from listed companies.

The regression results are shown in Table 6. Among them, the empirical results in columns (1) and (3) demonstrates that the digital transformation among upstream and downstream enterprises significantly drive the digital transformation within this industry. The observations in columns (2) and (4) suggest that such digital transformation reduce the carbon emission intensity of the enterprises. The digital transformation of upstream and downstream enter-prises not only mitigates the carbon emission intensity through the technology spillover but also fosters carbon emission reduction through the non-technology spillover associated with the intermediate goods market. This is attributable to the enhanced digital transformation in

**Table 6. Analysis of industry linkage effects.**

| variable | Backward (downstream) | | Forward (upstream) | |
|---|---|---|---|---|
| | (1) | (2) | (3) | (4) |
| | dig | CEI | dig | CEI |
| dig_down | 0.008** | -0.034*** | | |
| | [0.003] | [0.010] | | |
| dig_up | | | 0.086* | -0.243** |
| | | | [0.048] | [0.105] |
| Control variables | YES | YES | YES | YES |
| Year FE | YES | YES | YES | YES |
| Individual FE | YES | YES | YES | YES |
| City FE | YES | YES | YES | YES |
| Industry FE | YES | YES | YES | YES |
| N | 5236 | 5236 | 2022 | 2022 |
| $R^2$ | 0.694 | 0.369 | 0.732 | 0.371 |

Note

***, ** and * indicate significance level of 1%, 5%and 10%, respectively.

the industrial chain, which not only strengthens technological exchanges and information-sharing capacity among enterprises, forming positive feedback of technological spillover, but also encourages a shift toward a low-carbon and green production mode. This facilitates the adoption of clean production technologies, and generates a positive industry spillover effect within the industrial chain and supply chain, ultimately reducing overall carbon intensity.

Next, we explored whether the peer effect of the digital transformation of listed companies influences carbon emission reduction performance. Given that firms within the same industry share similar market positions, technological environments, and development prospects, and often engage in competitive relationships [58], does the digital transformation of industry trigger imitation behavior and create a transmission mechanism for carbon emission reduction? The above relationship not only exists between the competitors; if the competitors are in the same region, they often encounter similar resource allocations and business environment [59], raising the question of whether regional mimicry effects of competitors in the region and transmission mechanisms for carbon emission reduction are present. To identify whether there is a peer effect in the digital transformation of listed companies, the following model is set up:

$$dig_{it} = \beta_1 industry\_dig_{it} + \beta X_{it} + \alpha_i + \theta_t + \mu_{it} \tag{8}$$

$$dig_{it} = \beta_1 region\_dig_{it} + \beta X_{it} + \alpha_i + \theta_t + \mu_{it} \tag{9}$$

In Eq (9) **industry_dig** is the degree of digital transformation of enterprises in the same industry cohort, and in Eq (10) **region_dig** is the degree of digital transformation of enterprises in the same region cohort. Meanwhile, drawing on the treatment of Yang et al. (2020) [60], both time and region effects are controlled in the regression of Eq (9), with time and industry effects controlled in the regression of Eq (10).

In order to test the influence of the peer effect of digital transformation on the carbon emission reduction performance of listed companies, the following regression model is set:

$$CEI_{it} = \beta_1 industry\_dig_{it} + \beta X_{it} + \alpha_i + \theta_t + \mu_{it} \tag{10}$$

**Table 7. Analysis of peer effects.**

| variable | (1) | (2) | (3) | (4) |
|---|---|---|---|---|
| | dig | CEI | dig | CEI |
| industry_dig | 0.393*** | -0.492** | | |
| | [0.027] | [0.182] | | |
| region_dig | | | 0.738*** | -0.582** |
| | | | [0.070] | [0.272] |
| Control variables | YES | YES | YES | YES |
| Year FE | YES | YES | YES | YES |
| Individual FE | NO | NO | YES | YES |
| City FE | YES | YES | NO | NO |
| N | 26650 | 26650 | 26650 | 26650 |
| $R^2$ | 0.663 | 0.210 | 0.665 | 0.210 |

Note

***, ** and * indicate significance level of 1%, 5%and 10%, respectively.

$$CEI_{it} = \beta_1 region\_dig_{it} + \beta X_{it} + \alpha_i + \theta_t + \mu_{it} \tag{11}$$

The regression results are shown in Table 7, in which the regression results in columns (1) and (3) indicate that the digital transformation within the same industry and region can promote the digital transformation of the focal enterprise, confirming the presence of the digital transformation peer effect. Columns (2) and (4) further show that this cohort effect can significantly contribute to the reduction of carbon intensity of focal enterprises. This suggests that the peer effect of digital transformation extends beyond mere technical imitation and follow-up; it also promotes collaboration and information sharing among enterprises in areas such as product design, production process, logistics system and customer service. This broadens the application scenarios of digital technology, significantly improving the energy efficiency and facilitating a reduction of carbon emission intensity.

## Analysis of the heterogeneity

Given the multiple determinants influencing the carbon emission reduction performance of listed companies, this paper examines both internal and external environments in which the companies are located. It utilizes the approach proposed by Liu et al. (2023) [61] to categorize the overall samples in this paper according to variations in public attention, governmental environmental regulation, and the internal factor intensities in the companies. This classification allows for an assessment of the diverse effects of digital transformation on carbon emission intensity under different levels of public attention, governmental environmental regulation and internal factor intensities. The data are segmented and analyzed individually to test the heterogeneous impact of listed companies' digital transformation on carbon emission intensity across different scenarios of public concern, governmental environmental regulation and internal factor intensity of enterprises. In order to guarantee the reliability of the regression results, we adopt the methodological framework of Lv et al. (2022) [62], conducting a heterogeneity test in this paper is tested for differences in coefficients between groups, with results of the grouping test and the differences in coefficients between groups detailed in Table 8.

Firstly, considering the influence of public attention on the enterprise's pollution reduction and carbon reduction, we referenced the research of Dong et al. (2021) [63], setting

**Table 8. Heterogeneity test.**

| variable | Public attention | | Level of environmental regulation | | Factor intensity | | |
|---|---|---|---|---|---|---|---|
| | High | Low | High | Low | Capital intensive | skill-intensive | labor-intensive |
| | (1) | (2) | (3) | (4) | (5) | (6) | (7) |
| dig | -0.048** | -0.037*** | -1.309*** | -0.063 | -0.857** | -0.307 | -0.199** |
| | [0.020] | [0.009] | [0.351] | [0.047] | [0.387] | [0.206] | [0.074] |
| N | 7857 | 13975 | 8954 | 17574 | 11262 | 10088 | 9244 |
| Intergroup coefficient test | 0.27 | | 4.61** | | 9.15*** | | |
| $R^2$ | 0.236 | 0.185 | 0.292 | 0.233 | 0.243 | 0.045 | 0.412 |

Note

***, ** and * indicate significance level of 1%, 5%and 10%, respectively.

"environmental pollution" as the search keywords, extracting the Baidu search volume to gauge the average daily "environmental pollution" searches of each city, and subsequently categorize into high attention areas and low attention areas based on the median daily search figures annually. The estimated results, presented in columns (1) and (2) of Table 8, suggests that the digital transformation of enterprises in regions with higher democratic concern tends to more effectively reduce carbon intensity. However, the between-group coefficient test shows that the coefficients of the two groups reveal no significant disparity at the 1% statistical level, indicating that the increase in democratic concern at this stage does not significantly increase the enthusiasm of listed companies in China to reduce carbon emissions. This might stem from the absence of a robust monitoring mechanism for corporate carbon disclosure, leading to non-standardized and opaque emission reporting by some listed companies, which fails to form effective social supervision and public opinion pressure. Therefore, the public's concern for environmental pollution does not effectively compel enterprises to improve their social responsibility for low-carbon production, resulting in insignificant difference in the impact coefficients between regions with high public concern and regions with low public concern.

Secondly, the extent of environmental regulation imposed by the government onstitutes an important exogenous variable that affects the development of enterprises' decarbonization and greening. This reflects the local government's commitment to managing pollution emissions. Investigating the heterogeneous impacts of listed companies' digital transformation on carbon emission reduction under different environmental regulation backgrounds enables an analysis of listed companies' specific performance in digital transformation and carbon emission reduction. Following the methodology of Liu et al. (2023) [64], environmental regulation is measured by calculating the ratio of pollution control expenditures to the industrial output value of a region for the current year, and the median is taken to be divided into two groups of high and low to carry out a group regression. The results are shown in columns (3) and (4) of Table 8, indicate a significant difference in the coefficients between the two groups. Specifically, firms in regions with stringent environmental regulation regions demonstrate a more pronounced reduction in carbon intensity than those in areas with lax environmental regulation regions. Higher environmental regulations reflect the strength and determination of local governments to combat pollution. In the face of stricter environmental policies and regulations, companies will increase their R&D investment in low-carbon technologies, striving for more sustainable and efficient production methods. This result further extends the findings of Song et al. (2022) [38], demonstrating that firms operating under stricter regional environmental regulations experience a more pronounced incentive effect of digitization on green

technological innovation, as well as a greater impact of digital transformation on reducing carbon intensity.

Thirdly, drawing on the study of Zhao (2021) [65], this study segments the sample of listed companies into three types of capital-intensive, technology-intensive and labor-intensive based on the differences in factor intensity and regressed separately. The results, detailed in Table 8, reveal that digital transformation significantly mitigates carbon emission intensity across three types of enterprises, with statistical significance at the 5% level, and the coefficients between the groups are significantly different at the 1% statistical level, suggesting a hierarchy in responsiveness: capital-intensive > technology-intensive > labor-intensive. This may be explained by the fact that, compared with labor-intensive enterprises, capital-intensive and technology-intensive enterprises generally possess more robust financial and technological foundation. Such assets facilitate the effective deployment of digital technology, thereby enhancing the carbon emission reduction effect of the digital transformation on listed companies. This will be expected to be more conducive to the impact of digital transformation on carbon emission reduction of listed companies. This result, together with the study by Wang and Li (2024) [27], demonstrates that digital transformation plays a more prominent role in technology-intensive and capital-intensive enterprises.

## Conclusions and enlightenment

### Conclusions

At present, improving enterprises' energy efficiency and conserving energy are not only the most straightforward and cost-effective strategies to realize the goal of " dual carbon target ", but they also play a pivotal role in boosting enterprise competitiveness and realize large-scale carbon emission reductions. Consequently, this study examines the direct impact, effect mechanism, and driving effect of digital transformation on carbon emission intensity, utilizing data from Chinese A-share listed companies spanning from 2007 to 2021. The findings are as follows:

1. At this stage, the digital transformation of listed companies can effectively reduce their carbon emission intensity, and this conclusion is reaffirmed following an endogeneity test and robustness tests. Further, this paper explores the nonlinear impact between enterprise digital transformation and carbon intensity dynamically, and finds that there is a U-shaped relationship between digital transformation and enterprise carbon intensity that decreases and subsequently rises.

2. Environmental governance costs and green technology innovation are critical mechanisms through which enterprise digital transformation influences carbon intensity. Notably, both the relationship between enterprise digital transformation and environmental governance costs, as well as that between enterprise digital transformation and green technology innovation, exhibit an inverted U-shaped pattern. This observation supports the U-shaped relationship between enterprise digital transformation and carbon intensity.

3. The driving effect found that the digital transformation of enterprises can significantly reduce the carbon emission intensity of upstream and downstream enterprises and enterprises in the same industry and the same region, through the industrial linkage effect and the cohort effect.

4. Heterogeneity analyses show that stringent environmental regulations enhance the effectiveness of digital transformation in reducing carbon emissions. Digital transformation has a stronger carbon intensity reduction effect on capital-intensive and technology-intensive firms compared to labor-intensive firms.

These conclusions underscore the multi-faceted impacts of firms' digital transformation on carbon intensity, highlighting the critical role of environmental governance costs and green technology innovation, and the necessity of considering industry linkage effects, cohort effects, and heterogeneity of firms' internal and external environments in a comprehensive understanding of the impacts of firms' digital transformation on carbon intensity.

## Enlightenment

This paper elucidates theoretical framework and provides empirical substantiation, deepening the understanding of the impact of enterprise digital transformation on the carbon emission reduction. It offers pivotal insights for exploring the development path of enterprise green low-carbon transformation in China:

First, local governments should prioritize policy support for the digital transformation of local enterprises. Considering the stage effect of enterprise digital transformation on carbon intensity, the primary objective is to actively facilitate enterprise digital transformation to achieve an optimal threshold. This strategy requires an acceleration of information and digital infrastructure initiatives, including deploying 5G networks, industrial internet and big data centers, to enhance the support framework for enterprise digital transformation and expedite the emergence of a digitally empowered China. Furthermore, it is essential to integrate green principles in the process of enterprise digital transformation. Local governments need to intensify the oversight and evaluation of enterprises in the process of digital transformation, intensify the enforcement of environmental regulations, advocate for advanced production methodologies such as smart manufacturing and green manufacturing via media publicity, policy clarification, and exchanges of experiences. Additionally, it is important to raise the public's concern about environmental pollution, ensure financial and technological support for labor-intensive enterprises undergoing digital transformation, aiming to reduce carbon emissions from digitalization activities and foster a sustainable digital transformation paradigm.

Secondly, it is important to utilize the optimal threshold of enterprise digital transformation as a constraint to leverage the transmission effects of environmental governance costs and green technology innovation. On the one hand, in addressing the potential surge in costs associated with environmental governance during the early stage of digital transformation, the government could mitigate enterprise burdens by offering financial subsidies or tax relief policies, thereby smoothing transition in the initial stage of digital transformation. Establishing a special fund for digital transformation could spur investments in upgrading environmental protection equipment, applying digital technologies and refining management system. On the other hand, this approach advocates the digitalization, intensification and greening of local traditional industries. Enhancing investments in R&D investment in digital and green technologies and fostering enterprise-driven green technology innovations and demonstrations are vital. Through the establishment of green technology innovation incentives, provide R&D financial support, etc., to stimulate the enthusiasm of enterprises in green technology innovation, promote the transformation of technological innovation achievements into practical applications. In addition, it is necessary to focus on the alternative roles of green technological innovation and digital transformation in reducing the intensity of carbon emission, and strategically allocate innovation resources by combining the needs of different stages of digital transformation and green technological innovation, to avoid the waste of resources and duplication of construction.

Thirdly, the central government needs to strengthen its focus on the comprehensive landscape of the industrial chain to facilitate the interconnection and interoperability of digital

deployment, while advancing cross-regional digitalization and collaborative development. Given that companies are not isolated entities in the process of digital transformation and environmental economy activities, it is essential to encourage collaborative cooperation between upstream and downstream enterprises within the industrial chain and supply chain. This includes enhancing the integration of digital technology in the industrial chain and supply chain, promoting the establishment of a green supply chain management system, and supporting leading enterprises in spearheading initiatives related to supply chain integration and innovative low-carbon management. Such efforts are crucial for effectively harnessing the industrial synergy effect associated with the carbon emission reduction performance of digital transformation. Additionally, this approach fosters regional digital synergistic development, augments the interaction and exchange of enterprises in the same industry and region. It encourages the establishment of digital resource-sharing platforms across neighboring areas, thereby facilitating a dynamic flow of data elements and fostering in-depth exchanges of digital technologies. Ultimately, this strategy aims to promote the cohort effect of enterprises' digital transformations and amplify the spatial spillover effects of carbon emission reduction resulting from digital transformation.

## Supporting information

**S1 File.**
(ZIP)

## Acknowledgments

We would like to express our gratitude to Pengyu Yang for collecting and writing information on the green development of enterprises, Kejia Guo for her contribution to the theoretical review and revision, Jing Jia for organizing and analyzing data on listed companies, and Yulin Yin for providing research methods and organizing the article. We appreciate for editors and reviewers' warm work earnestly.

## Author Contributions

**Conceptualization:** Pengyu Yang.

**Funding acquisition:** Pengyu Yang.

**Investigation:** Pengyu Yang.

**Visualization:** Kejia Guo, Yulin Yin.

**Writing – original draft:** Kejia Guo, Jing Jia, Yulin Yin.

**Writing – review & editing:** Jing Jia.

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
