## [Decision Letter · Decision Letter 0]

5 Jun 2024

PONE-D-23-41742The Direct Role, Mechanism Analysis, and Driving Effect of Digital Empowerment of Carbon Reduction in Enterprises ——Empirical evidence from listed companiesPLOS ONE

Dear Dr. Jing,

Thank you for submitting your manuscript to PLOS ONE. After careful consideration, we feel that it has merit but does not fully meet PLOS ONE’s publication criteria as it currently stands. Therefore, we invite you to submit a revised version of the manuscript that addresses the points raised during the review process.

We look forward to receiving your revised manuscript.

Kind regards,

Xiaoyong Zhou, Ph.D.

Academic Editor

PLOS ONE

Reviewers' comments:

Reviewer's Responses to Questions

**Comments to the Author**

1. Is the manuscript technically sound, and do the data support the conclusions?

Reviewer #1: Yes

Reviewer #2: Yes

2. Has the statistical analysis been performed appropriately and rigorously? 

Reviewer #1: Yes

Reviewer #2: Yes

3. Have the authors made all data underlying the findings in their manuscript fully available?

Reviewer #1: No

Reviewer #2: No

4. Is the manuscript presented in an intelligible fashion and written in standard English?

Reviewer #1: Yes

Reviewer #2: Yes

5. Review Comments to the Author

Reviewer #1: This article studies the impact and mechanism of digital transformation of enterprises on carbon emissions. Empirical research verifies theoretical hypotheses. However, there are still the following issues that need further improvement.

1. In the theoretical analysis section, it is recommended to add a theoretical model framework diagram to facilitate readers to understand the author's research ideas.

2. It is recommended that the author compare the obtained results with research results in relevant fields and analyze the applicable scenarios of relevant theories based on similarities and differences.

3. The issue of collinearity in data needs to be tested.

4. Why is there such a significant difference in R2 in Table 2? The R2 of model (1) is 0, model (4) only added a "Regional effect" compared to model (3), but R2 increased from 0.295 to 0.92.

5.From the author's theoretical analysis and model design, reducing the cost of environmental governance, improving the competition degree of the industry, and increasing the green innovation are mediating effects in the impact of digital transformation on carbon emissions, but the author did not rigorously test these mediating effects, such as sobel and bootstrap tests.

Reviewer #2: This manuscript aims to comprehensively investigate the nexus between the digital transformation and carbon emission reduction using panel data of A-share listed companies in China (2007-2021). This topic is relevant and interesting for readers of PLOS ONE. The method is clearly explained, the result report is logical and the conclusion is supported by the data. In presentation, however, the organization and demonstration of this manuscript can be improved. Therefore, I suggest to accept this manuscript for publication pending some minor revisions. Recommendations for improvement are listed below.

(1) Abstract: The authors should clearly and systematically present the direct, mediating, driving, and heterogeneous effects of corporate digital transformation on carbon emissions reduction, ensuring coherence from beginning to end.

(2) Literature review and research hypothesis: Given the numerous variables and mechanism pathways discussed in the article, it is recommended to include a framework diagram to elucidate the specific impact mechanisms and pathways.

(3) Data source and description: The financial sector is generally excluded, at least during the sample selection process. Given the unbalanced panel data, it is necessary to indicate the number of listed companies throughout the sample period.

(4) Benchmark regression: Significantly positive quadratic terms of explanatory variable may not necessarily be a U-shaped relationship, and the U-shaped curve can be further U-tested by referring to the literature " Thinking about U: Theorizing and testing U- and inverted U-shaped relationships in strategy Research ".

(5) Endogeneity and robustness test: The construction of instrumental variables yields only cross-sectional data, requiring further explanation on how they are treated as panel data. Additionally, addressing the endogeneity problem with double differencing is commendable, but the parallel trend assumption is a prerequisite for its validity. Therefore, it is advisable to conduct and pass the parallel trend test.

(6) Analysis of the heterogeneity: The regression coefficients of different subgroups are significant and in the same direction, and cannot be simply compared to the size of the coefficients, which needs to be tested by the coefficient of variation between groups.

(7) There are some grammar mistakes in this manuscript. Please ask some native English writers to help you improve this paper.

6. PLOS authors have the option to publish the peer review history of their article (what does this mean?). If published, this will include your full peer review and any attached files.

Reviewer #1: **Yes: **Hongjun Guan

Reviewer #2: No

---

## [Author Response · Author response to Decision Letter 0]

26 Jul 2024

We are very grateful for your constructive comments and suggestions for our manuscript. we have considered the comments very carefully and have revised the paper accordingly. These comments are all valuable and helpful for improving our article. We have extensively modified our manuscript according to the editor and reviewers’ comments. We hope that the corrections are satisfactory. Point-by-point answers to the kind editor and two kind reviewers are listed in"Response to Reviewers.docx" . 

The reviewer comments are in italic font below, and specific concerns have been numbered. Our response is given in regular font, and changes/additions to the manuscript are shown in blue text. We have tried our best to make all the revisions clear, and we hope that the revised manuscript can satisfy the requirements for publication. We appreciate for editors and reviewers’ warm work earnestly, and hope that the correction will meet with approval.

---

## [Decision Letter · Decision Letter 1]

9 Sep 2024

PONE-D-23-41742R1Corporate digital transformation and Carbon Emission Intensity: Empirical evidence from listed companies in ChinaPLOS ONE

Dear Dr. Guo,

Thank you for submitting your manuscript to PLOS ONE. After careful consideration, we feel that it has merit but does not fully meet PLOS ONE’s publication criteria as it currently stands. Therefore, we invite you to submit a revised version of the manuscript that addresses the points raised during the review process.

We look forward to receiving your revised manuscript.

Kind regards,

Tianheng Shu, PhD

Academic Editor

PLOS ONE

Journal Requirements:

Reviewers' comments:

Reviewer's Responses to Questions

**Comments to the Author**

1. If the authors have adequately addressed your comments raised in a previous round of review and you feel that this manuscript is now acceptable for publication, you may indicate that here to bypass the “Comments to the Author” section, enter your conflict of interest statement in the “Confidential to Editor” section, and submit your "Accept" recommendation.

Reviewer #2: (No Response)

Reviewer #3: (No Response)

2. Is the manuscript technically sound, and do the data support the conclusions?

Reviewer #2: Yes

Reviewer #3: Yes

3. Has the statistical analysis been performed appropriately and rigorously? 

Reviewer #2: Yes

Reviewer #3: Yes

4. Have the authors made all data underlying the findings in their manuscript fully available?

Reviewer #2: No

Reviewer #3: (No Response)

5. Is the manuscript presented in an intelligible fashion and written in standard English?

Reviewer #2: Yes

Reviewer #3: Yes

6. Review Comments to the Author

Reviewer #2: The authors have addressed most of the reviewers’ comments appropriately, resulting in an improved quality of the paper. However, several minor comments/suggestions remain:

1. The introduction section remains insufficient. It is recommended that after introducing the topic, the section should include a description of the research strategy and clearly indicate the marginal contribution of this paper compared to existing studies.

2. The conclusion should succinctly summarize the core findings of the study. The comparative analysis of the results can be discussed after the empirical results in the previous section.

3. This paper reveals a U-shaped relationship between digitalization and carbon intensity of enterprises, providing new perspectives for policymakers. The authors can refer to “U-shaped relationship between digitalization and low-carbon economy: Mediation and spillover effects, JCLP, 2024” to enhance the corresponding policy implications.

4. Further improvement of the reference format is recommended.

Reviewer #3: Dear Authors, I hope this feedback finds you well. I have carefully reviewed your paper and would like to provide some suggestions for improvement. Please consider the following points.

1. The current introduction is a bit too brief, and it's suggested that the author add three more paragraphs. The first and second paragraphs stay the same. The third paragraph should be transformed into the main research content and conclusion of this paper. The fourth paragraph should be modified to the research contribution (with the contribution in the literature review removed), and the fifth paragraph should be converted to an introduction of the structure of the paper. It's important to note that when writing the research contribution, the author needs to cite previous literature. In other words, summarize in one sentence what previous researchers have done, what problems still exist, and how your research resolves these problems.

2. The rest of the paper is fine. I'd like to offer the author a tip. When uploading the revised manuscript, please don't upload the revised mode manuscript. Just highlight the modified parts.

7. PLOS authors have the option to publish the peer review history of their article (what does this mean?). If published, this will include your full peer review and any attached files.

Reviewer #2: No

Reviewer #3: No

---

## [Author Response · Author response to Decision Letter 1]

8 Oct 2024

Dear editors and reviewers,

Thank you very much for giving us an opportunity to revise our manuscript. We are very grateful for your constructive comments and suggestions for our manuscript. Those comments are all valuable and very helpful for revising and improving our paper. We have considered the comments very carefully and have revised the paper accordingly. Based on the comments of the two reviewers, we mainly revised the introductory and conclusions and implications sections of the article and improved the references format according to the style of the journal formatting specifications and requirements. All revised contents are marked in red in the revised manuscript with track changes.Point-by-point answers to two kind reviewers are listed in "Response to Reviewers.docx".

We sincerely appreciate the time and effort invested by editors and reviewers in evaluating our manuscript, and hope that the correction will meet with approval.

---

## [Decision Letter · Decision Letter 2]

1 Nov 2024

Corporate digital transformation and Carbon Emission Intensity: Empirical evidence from listed companies in China

PONE-D-23-41742R2

Dear Dr. Guo,

We’re pleased to inform you that your manuscript has been judged scientifically suitable for publication and will be formally accepted for publication once it meets all outstanding technical requirements.

Kind regards,

Tianheng Shu, PhD

Academic Editor

PLOS ONE

Additional Editor Comments (optional):

Reviewers' comments:

Reviewer's Responses to Questions

**Comments to the Author**

1. If the authors have adequately addressed your comments raised in a previous round of review and you feel that this manuscript is now acceptable for publication, you may indicate that here to bypass the “Comments to the Author” section, enter your conflict of interest statement in the “Confidential to Editor” section, and submit your "Accept" recommendation.

Reviewer #2: (No Response)

2. Is the manuscript technically sound, and do the data support the conclusions?

Reviewer #2: Yes

3. Has the statistical analysis been performed appropriately and rigorously? 

Reviewer #2: Yes

4. Have the authors made all data underlying the findings in their manuscript fully available?

Reviewer #2: No

5. Is the manuscript presented in an intelligible fashion and written in standard English?

Reviewer #2: Yes

6. Review Comments to the Author

Reviewer #2: The author has revised the corresponding parts as required, and the article as a whole seems to be of a certain level and has reached the level of publication.

7. PLOS authors have the option to publish the peer review history of their article (what does this mean?). If published, this will include your full peer review and any attached files.

Reviewer #2: No

---

## [Editor Report · Acceptance letter]

8 Dec 2024

PONE-D-23-41742R2 

PLOS ONE

Dear Dr. Guo, 

I'm pleased to inform you that your manuscript has been deemed suitable for publication in PLOS ONE. Congratulations! Your manuscript is now being handed over to our production team.

Kind regards, 

on behalf of

Dr. Tianheng Shu 

Academic Editor

PLOS ONE